# Molecular and Environmental Modulators of Aging: Interplay Between Inflammation, Epigenetics, and RNA Stability

**DOI:** 10.3390/genes16070796

**Published:** 2025-07-01

**Authors:** Konstantina Dragoumani, Dimitris Kletsas, George P. Chrousos, Dimitrios Vlachakis, Nikolaos A. A. Balatsos

**Affiliations:** 1Department of Biochemistry and Biotechnology, University of Thessaly, Biopolis, 415 00 Larissa, Greece; kdragoumani@uth.gr; 2Laboratory of Cell Proliferation & Ageing, Institute of Biosciences & Applications, National Centre for Scientific Research ‘Demokritos’, 153 10 Agia Paraskevi, Greece; dkletsas@bio.demokritos.gr; 3University Research Institute of Maternal and Child Health and Precision Medicine, School of Medicine, National and Kapodistrian University of Athens, 115 27 Athens, Greece; chrousge@med.uoa.gr; 4Laboratory of Genetics, Department of Biotechnology, School of Applied Biology and Biotechnology, Agricultural University of Athens, 118 55 Athens, Greece; 5Algorithms and Bioinformatics Group, Informatics Department, Faculty of Natural, Mathematical & Engineering Sciences, Strand Campus, King’s College London, London WC2R 2LS, UK

**Keywords:** aging, longevity, cellular senescence, telomere dynamics, aging pathways, *Klotho* gene, ACE, NF-κB inflammation

## Abstract

Aging is a complex biological process characterized by the progressive accumulation of cellular and molecular damage, leading to functional decline and increased susceptibility to age-related diseases. Central to this process is cellular senescence, a state of irreversible cell cycle arrest that acts as both a protective mechanism against tumorigenesis and a contributor to tissue degeneration. Herein, we explore the genetic and molecular mechanisms underlying aging, with a focus on telomere dynamics, the *Klotho* gene, angiotensin-converting enzyme (ACE), and the NF-κB pathway. Telomeres, which serve as protective caps at chromosome ends, shorten with each cell division, leading to replicative senescence, while the enzyme telomerase plays a pivotal role in maintaining telomere length and cellular longevity. The *Klotho* gene encoding for an aging suppressor influences insulin/IGF-1 signaling and has antioxidant properties that protect against oxidative stress. ACE, through its dual role in regulating blood pressure and degrading amyloid-beta, impacts longevity and age-related pathologies. The NF-κB pathway drives chronic inflammation or “inflammaging,” contributing to the onset of age-related diseases. Understanding these pathways offers promising avenues for therapeutic interventions to extend health span and lifespan. Targeting mechanisms such as telomerase activation, *Klotho* supplementation, ACE inhibition, and NF-κB modulation hold potential for combating the detrimental effects of aging and promoting healthier aging in the population.

## 1. Introduction

Aging is a complex biological process resulting from the accumulation of damage at the cellular and molecular levels over time. This gradual decline leads to reduced functionality and an increased susceptibility to diseases, ultimately contributing to mortality. Cellular senescence, a state of irreversible cell cycle arrest, is one of the central hallmarks of aging, acting as both a protective mechanism against tumorigenesis and a contributor to age-related degeneration [1]. The genetics of aging encompasses a wide array of factors, from telomere dynamics to key regulatory genes like *Klotho* and *ACE*, which play crucial roles in the aging process and longevity. Recent advances in molecular biology and genetics have revealed several pathways and mechanisms that influence cellular aging, offering promising targets for interventions aimed at promoting health span and extending lifespan. Understanding the mechanisms governing cellular senescence offers valuable insights into the aging process and may pave the way for therapeutic interventions to promote healthy aging and longevity.

## 2. Hallmarks of Aging

As senescence represents the most substantial undertaking in the human life course, it is imperative to undertake a thorough examination and provide a precise definition. Several interconnected biological and physiological processes have been identified as hallmarks (Table 1), collectively contributing to the natural and inevitable process of gradual decline in physiological function, which is known as aging and the onset of age-related diseases. The hallmarks can be categorized as follows:

The genomic instability represents the progressive accumulation of DNA damage, including point mutations, chromosomal aberrations, and impaired DNA repair mechanisms. All these phenomena accompany the aging process and arise due to oxidative stress, replication errors, and environmental factors such as radiation and toxins [2]. It consists of an overall genomic state that characterizes the aged organism. The resulting transcriptional noise alters cellular function, increasing heterogeneity among aging cells [3]. Furthermore, genomic instability has been implicated in tumorigenesis, contributing to the clonal expansion of cells harboring oncogenic mutations [4].

Upon DNA replication, a section of the telomeric DNA is lost, a process known resulting to telomere shortening or telomere attrition. It is an additional distinct genomic state noted during the aging process. The occurrence of critically short telomeres instigates DNA damage responses (DDRs), consequently resulting in cellular senescence or apoptosis. Telomere erosion has been found to be associated with histone modifications and epigenetic alterations, which exacerbate genomic instability and contribute to aging-related dysfunctions [5,6,7].

Furthermore, aging is characterized by widespread epigenetic alterations, including DNA methylation shifts, histone modifications, and chromatin remodeling [8]. The progressive loss of DNA methylation in repetitive elements contributes to genomic instability, whereas hypermethylation at promoter CpG islands leads to gene silencing. The development of epigenetic clocks, such as the Horvath clock, has provided insights into biological aging rates. Consequently, changes in histone modification patterns have been observed in correlation with age progression, with implications for gene expression and the structure of the chromatin. These changes, in turn, can affect cellular function and, thus, cellular health and state along the way.

In terms of cellular functional disruption, loss of proteostasis has been found at high prevalence during the aging process. Proteostasis is defined as the set of cellular mechanisms that regulate protein synthesis, folding, and degradation. Dysregulation of proteostasis in aged cells has been demonstrated to result in the accumulation of misfolded proteins, thus promoting the development of neurodegenerative diseases such as Alzheimer’s and Parkinson’s [9]. Age-related declines in autophagy and chaperone-mediated protein refolding exacerbate proteotoxic stress, contributing to cellular dysfunction and tissue degeneration [10].

Moreover, mitochondria are pivotal to cellular energy production, and their dysfunction is a hallmark of aging. During senescence, mitochondria experience a decline in function, characterized by mitochondrial DNA (mtDNA) mutations, increased oxidative stress, damage-associated molecular patterns (DAMPs) formation, decreased mitochondrial energy conversion, and hampered mitochondrial turnover. Accumulation of mtDNA mutations, increased reactive oxygen species (ROS) production, and impaired oxidative phosphorylation contribute to metabolic decline [11]. Furthermore, mitochondrial dysfunction has been implicated in the development of inflammation and immune dysregulation, which in turn accelerates the aging process [12]. Targeting mitochondrial homeostasis through pharmacological or genetic interventions may mitigate aging-related decline.

The actual cellular senescence presented in the aging process is defined as a state of irreversible cell cycle arrest, which is triggered by telomere attrition, DNA damage, and oncogenic stress [13]. The senescence-associated secretory phenotype (SASP) refers to the collective secretion of inflammatory cytokines, growth factors, and proteases by senescent cells. Key inflammatory cytokines involved include interleukin-6 (IL-6), interleukin-1α/β (IL-1α/β), tumor necrosis factor-alpha (TNF-α), and chemokines such as IL-8 (CXCL8) and monocyte chemoattractant protein-1 (MCP-1/CCL2). Prominent growth factors include vascular endothelial growth factor (VEGF), transforming growth factor-beta (TGF-β), epidermal growth factor (EGF), and fibroblast growth factors (FGFs). While transient senescence can play a beneficial role in tissue repair, the chronic accumulation of senescent cells contributes to sustained inflammation, impaired tissue function, and the progression of age-related diseases.

Furthermore, aging is associated with a decline in stem cell function and regenerative potential. The exhaustion of stem cells has been demonstrated to result in impaired tissue homeostasis, characterized by diminished cellular turnover and compromised repair mechanisms [14]. Hematopoietic stem cells, mesenchymal stem cells, and neural stem cells exhibit diminished proliferative capacity with age, contributing to immune dysfunction, osteoporosis, and cognitive decline, respectively.

Another defining feature of cellular aging is altered intercellular communication. Aging has been demonstrated to disrupt systemic communication networks, leading to chronic inflammation and impaired tissue homeostasis. Increased pro-inflammatory signaling, often termed “inflammaging,” contributes to immune system dysfunction and the progression of age-related diseases such as cardiovascular disease and neurodegeneration. Furthermore, the exacerbation of cellular aging is compounded by dysregulated hormonal signaling and extracellular matrix remodeling [15].

Aging is also associated with a persistent, low-grade inflammatory state known as inflammaging, characterized by the continuous secretion of pro-inflammatory cytokines, chemokines, and other signaling molecules by aged and senescent cells. This chronic, low-grade inflammation can inhibit the immune system and potentially cause muscle wasting, bone loss, and other harmful effects. Inflammaging disrupts tissue homeostasis and contributes to the development of age-related diseases, including neurodegenerative disorders such as Alzheimer’s disease, cardiovascular diseases, and metabolic disorders. Despite its detrimental effects, inflammaging may have evolved as a trade-off, providing beneficial immune responses in early life while becoming maladaptive with age [16,17].

Age-related dysbiosis is another significant phenomenon associated with alterations in the gut microbiome. This shift is characterized by a decline in microbial diversity, a reduction in beneficial bacteria (such as *Bifidobacteria* and *Lactobacilli*), and an increase in pro-inflammatory and pathogenic species. These microbial changes contribute to chronic low-grade inflammation, or inflammaging, by disrupting gut barrier integrity and promoting systemic immune activation. Dysbiosis has been linked to various age-related conditions, including metabolic disorders, neurodegenerative diseases, and impaired immune function. Factors such as dietary changes, medications (e.g., antibiotics and proton pump inhibitors), and reduced physical activity further exacerbate microbiome imbalances in older individuals [18,19].

These hallmarks interact synergistically, reinforcing each other to drive the aging process, and together they create the senescence landscape. A comprehensive understanding of these interconnected mechanisms provides a foundation for the development of targeted therapeutic strategies to promote healthy aging and extend lifespan.

**Table 1 genes-16-00796-t001:** Hallmarks of aging.

	Hallmark	Effect	Indicative References
1	Cellular senescence/irreversible cell cycle arrest	Telomere attrition DNA damage Oncogenic stress Collective secretion of inflammatory cytokines, growth factors, and proteases	[13,20,21]
2	Genomic instability	Point mutations Chromosomal aberrations Impaired DNA repair	[2,22,23]
3	Telomere attrition	Telomere erosion Histone modifications Epigenetic alterations	[7,24,25]
4	Epigenetic alterations	DNA methylation shifts Histone modifications Chromatin remodeling	[8,26,27]
5	Loss of proteostasis	Proteotoxic stress Cellular dysfunction Tissue degeneration	[10,28,29]
6	Mitochondrial dysfunction	mtDNA mutations Increased oxidative stress, DAMP formation Decreased mitochondrial energy conversion Hampered mitochondrial turnover dynamic	[11,30,31]
7	Stem cell function decline	Impaired tissue homeostasis Diminished cellular turnover Compromised repair mechanisms Diminished proliferative capacity Immune dysfunction Osteoporosis Cognitive decline	[14,32,33]
8	Altered Intercellular Communication	Chronic inflammation Impaired tissue homeostasis Immune system dysfunction Dysregulated hormonal signaling Extracellular matrix remodeling	[15,17,34]
9	Chronic inflammation	Continuous secretion of pro-inflammatory cytokines, chemokines, and other signaling molecules Inhibition of the immune system Disruption of tissue homeostasis Development of age-related diseases	[16,17,35,36]
10	Dysbiosis	Chronic inflammation Impaired immune function Metabolic disorders Neurodegenerative diseases	[18,19,37,38]

## 3. Cellular Senescence

Cellular senescence is a stress response that entails an irreversible cessation of mitotic activity. As such, the senescence response is a potent tumor suppressive mechanism but has also been implicated in the loss of physiological functions and increased disease risk associated with aging.

Cellular senescence represents a dynamic and multifaceted process triggered by various intrinsic and extrinsic stressors. The initiation of senescence typically occurs in response to stress signals such as DNA damage, oxidative stress, oncogenic signaling, and telomere shortening [39]. These signals activate tumor suppressor pathways, most notably the p53/p21 and p16^INK4a^/Rb pathways, leading to the arrest of cell proliferation. The cell cycle arrest in senescence is not merely a cessation of cell division; it involves extensive chromatin remodeling, metabolic alterations, and secretion of bioactive molecules known as the senescence-associated SASP. The formation of heterochromatin structures, known as senescence-associated heterochromatic foci (SAHF), contributes to the silencing of proliferation-promoting genes, reinforcing the senescence state. Moreover, increased expression of cell cycle inhibitors like p16^INK4a^ and p21 ensures that cells remain in a state of permanent arrest. The SASP is a hallmark of full senescence and consists of a diverse range of cytokines, chemokines, growth factors, and matrix metalloproteinases that can have both autocrine and paracrine effects [40]. While SASP factors can reinforce the senescence state and contribute to tissue remodeling and repair, they can also drive chronic inflammation, known as “inflammaging,” and promote the progression of age-related diseases [41]. The discovery of the SASP has led to the development of senolytics, a new class of small molecules that selectively target and eliminate senescent cells, offering potential for delaying or preventing age-related diseases. These compounds have shown promise in human studies.

Interestingly, the composition of the SASP is highly context-dependent, varying based on the type of senescent cell, the tissue environment, and the nature of the inducing stressor. Research has identified key regulatory pathways that influence the development and maintenance of the SASP, including the NF-κB, mTOR, and p38 MAPK signaling pathways. The involvement of these pathways suggests that cellular senescence is not merely a passive process but rather a highly regulated and dynamic state. The mTOR pathway, for instance, plays a crucial role in regulating protein synthesis, autophagy, and metabolic activity in senescent cells, linking nutrient-sensing pathways to aging [42].

Late senescence is characterized by the persistence of senescent cells, which exhibit resistance to apoptosis. This phase can have detrimental effects on tissue function due to the prolonged secretion of SASP factors and the induction of chronic inflammation. Acute senescence typically serves as a beneficial, transient response to injury, aiding in wound healing and tissue regeneration. In contrast, chronic senescence, which accumulates with age, contributes to tissue dysfunction, fibrosis, and the progression of age-related diseases such as osteoarthritis, atherosclerosis, and neurodegeneration [43]. The clearance of senescent cells by the immune system is an essential aspect of maintaining tissue homeostasis, and the failure of immune surveillance with age results in the accumulation of these cells. Strategies such as senolytics, which are drugs designed to selectively eliminate senescent cells, have shown promise in extending health span and ameliorating age-related pathologies in preclinical models [44].

Among the inducers of cellular senescence, particularly in human cells, is the telomere attrition that conveys repeated cell division in the absence of telomerase, as well as other forms of DNA damage, most notably DNA double-strand breaks and oxidative stress (OS)1 [13]. Due to the end-replication problem first proposed by Olovnikov [45] and later by Watson [46], telomere length naturally erodes with each cell division, leading to progressive shortening with age. When telomeres reach a critically short length, they lose the ability to provide end-protection and to prevent DDRs, resulting in a permanent cell cycle arrest known as replicative senescence. In normal cells, this senescence acts as a barrier to unlimited cell growth and replicative immortality—a key hallmark of cancer—thus serving as a vital tumor suppressor mechanism.

## 4. Genomic Instability

Genomic instability is a critical phenomenon characterized by an increased frequency of mutations within the genome, which encompasses various genetic alterations such as point mutations, insertions, deletions, chromosomal rearrangements, and aneuploidy. These alterations can arise from both intrinsic factors, such as DNA replication errors and the production of reactive oxygen species (ROS), and extrinsic factors, including exposure to environmental toxins and radiation [14]. The cellular mechanisms responsible for maintaining genomic integrity, including DNA repair pathways, cell cycle checkpoints, and apoptosis, tend to become less efficient with age. This decline in genomic maintenance capacity facilitates the accumulation of genetic damage, thereby contributing to heightened genomic instability [14] (Table 2).

The accumulation of genomic alterations leads to an increased mutation load and genetic heterogeneity, often referred to as “somatic mosaicism,” within dividing cellular populations [2,13,47,48,49,50,51,52,53]. This phenomenon can impair tissue function and overall organismal health. The extent of mutation accumulation with age varies significantly across different organs and cell types, with tissues that exhibit high rates of cell turnover—such as skin, liver, and hematopoietic cells—being particularly affected [2,47]. The bone marrow serves as an exemplary organ system for studying the interplay between genomic instability and genetic heterogeneity due to its rich source of hematopoietic stem cells (HSCs) and progenitor cells [47,54,55]. As organisms age, the accumulation of genetic mutations within the bone marrow contributes to an increasing mutation load, resulting in significant genetic diversity among cellular populations [2]. This clonal diversity can influence cellular behaviors, with some clones potentially gaining a proliferative advantage or resistance to apoptosis, thereby exacerbating the effects of genomic instability.

The implications of genomic instability and the resultant genetic heterogeneity are profound, particularly concerning tissue function and aging [2,47,56,57,58,59]. As genetically diverse clones expand, they can disrupt normal tissue architecture and function. For instance, conditions, such as Clonal hematopoiesis of indeterminate potential (CHIP), characterized by the clonal expansion of mutated hematopoietic stem cells [14], and Monoclonal Gammopathy of Undetermined Significance (MGUS), can impair hematopoiesis in the bone marrow, leading to reduced production of healthy blood cells and an increased risk of hematological malignancies. These conditions are associated with heightened incidences of cardiovascular aging phenotypes, including an increased risk of thrombotic events and cardiovascular diseases [60,61,62,63,64,65,66,67,68,69]. Common mutations in CHIP occur in epigenetic regulators, such as DNMT3A and TET2, which lead to alterations in DNA methylation. Loss of function in DNMT3A or TET2 is associated with increased genomic instability and impaired DNA repair mechanisms. Moreover, the prevalence of CHIP increases with age, particularly after 50 years. In line with this, genomic instability can also induce cellular senescence, a state of irreversible growth arrest that contributes to the depletion of stem cell pools and impairs tissue regeneration [70]. Collectively, these processes accelerate the functional decline of tissues, manifesting as the phenotypic hallmarks of aging and increasing susceptibility to age-related diseases.

In addition to its role in aging, genomic instability is a well-established contributor to various diseases, particularly cancer. Somatic genomic instability is believed to drive tumorigenesis and is a hallmark of the substantial heterogeneity observed among different cancer types [15,71,72]. The ability of cancer cells to alter their genetic makeup over time is a significant factor in poor prognosis and therapy resistance. When genomic instability occurs in germ cells, it can lead to heritable genetic alterations, contributing to human genetic diseases. The accumulation of mutations, including genome rearrangements and aneuploidy, has been observed to increase with age in various tissues, such as the liver and brain, although the patterns of accumulation can differ significantly between these organs. Understanding the mechanisms by which genomic instability contributes to aging and disease is essential for developing strategies aiming to mitigate its effects and promote healthy aging [4].

Genome rearrangements, including translocations and large deletions, are a major component of the mutation spectrum in certain tissues at old age, such as the heart [3]. A process of gradual genome alterations could increase gene transcriptional noise, thereby altering the effectiveness of the cell’s network of functional pathways. Apart from cancer, which is a clonal disease, few disease endpoints have been considered as potential outcomes of mutation accumulation in somatic cells. While mutation accumulation has been implicated as a general cause of aging, no specific mechanism has been proposed to explain how it leads to the multitude of degenerative processes that comprise aging.

Although only a few genome rearrangements were detected with age in the brain, other studies have reported an age-related increase in the loss of heterozygosity events, a type of mutation that cannot be detected using the lacZ system [73,74,75]. The latter has been widely applied in genotoxicity assays to detect mutations in various tissues, including the brain, liver, and lungs, by inserting the β-galactosidase gene (*lacZ*) into the genome of transgenic mice. When mutations occur in *lacZ*, they can be identified through changes in β-galactosidase activity. Additionally, aneuploidy in aging mouse neuronal cells, another type of genomic alteration not detectable by the lacZ system, appears to be widespread in the brain [69]. Chromatin alterations, such as changes in DNA methylation or modifications of the histone code, could also increase transcriptional noise in postmitotic cells. A more detailed investigation into the mechanisms by which different postmitotic cell types generate increased transcriptional noise through random genome alterations could provide new insights into the role of genotoxic stress in aging and disease.

In the liver of mice, mutant frequencies increase with age from birth to 34 months, while in the brain, an increase is observed only between birth and 4–6 months [53]. Molecular characterization of these mutations revealed that a significant proportion involved genome rearrangement events, with one breakpoint in a reporter gene and the other in the mouse flanking sequence. In the liver, genome rearrangements did not increase with age until after 27 months, when they grew rapidly. In contrast, the frequency of genome rearrangements in the brain was lower than in the liver and did not increase with age. It has been suggested that oxidative stress is a key factor for the rapid rearrangements observed in the liver of old mice; oxidative DNA damage has been documented in the liver, while in the brain, it is absent or less evident [76].

Nevertheless, more recently, DNA damage contributes to brain aging and neurodegenerative diseases. Among the factors examined, histone deacetylase 1 (HDAC1) expression is minimal in both Alzheimer patients and normally aging adults, and HDAC1-deficient mice accumulate age-associated DNA damage and cognitive impairment. HDAC1 has been involved in OGG1-initiated 8-oxoguanine (8-oxoG) repair in the brain. Pharmacological activation of HDAC1 alleviated the effects of 8-oxoG in aged mice, highlighting the therapeutic potential of the enzyme in brain aging and neurodegeneration [77].

## 5. Telomere Attrition and Telomerase in Aging

Telomeres are nucleoprotein complexes that cap the ends of linear chromosomes. They are composed of highly conserved, tandem arrays of G-rich repetitive sequences (5′-TTAGGG-3′ in humans and other vertebrates) [7] and serve as protective caps that prevent genomic instability [78]. They terminate in a G-rich 3′ single-stranded overhang, with both double-stranded and single-stranded telomeric regions bound by proteins collectively termed shelterin. Shelterin protects chromosomal termini from degradation and fusion, preventing natural chromosome ends from being mistakenly recognized as DNA double-strand breaks (DSBs), which could trigger deleterious DNA damage responses (DDRs).

Telomeres are synthesized by telomerase, a ribonucleoprotein complex, which plays a pivotal role in counteracting telomere shortening [79]. Two essential components of human telomerase are telomerase RNA component (hTERC) and telomerase reverse transcriptase (hTERT), which synthesize telomere repeats. While telomerase activity is primarily repressed in most somatic cells, it remains active in stem cells and certain immune cells, contributing to cellular longevity. Recent studies suggest that telomerase exhibits dual functions: a canonical role in telomere maintenance and a non-canonical role in mitochondrial function, thus influencing cellular aging [80]. The regulation of telomerase and telomere dynamics has profound implications for aging and age-related diseases. For example, in endothelial cells, inhibition of telomerase activity induces cellular senescence and contributes to vascular dysfunction [81]. Moreover, emerging evidence indicates that telomerase interacts with telomeres in two distinct manners: short, dynamic interactions that probe telomere lengths and long, stable interactions that enable telomere elongation [82].

Telomere shortening is linked to global reductions in histone levels and epigenetic changes, which contribute to aging and age-related diseases, such as DNA methylation and biological clocks. Emerging evidence suggests an interplay between telomere attrition and metabolic imbalance or mitochondrial dysregulation, consistent with reports of a positive association between mitochondrial DNA and telomere length, indicating co-regulation of telomeres and mitochondrial function.

Many studies, including Sanchez et al., 2024, have demonstrated that mean human telomere length decreases with age [83]. In this study, the authors deployed digital telomere measurement (DTM) to reveal any association between the distributions of telomere length with age and disease in humans. Telomere length distributions from a cross-section of 63 healthy and diseased human samples were examined, and the findings suggest that the structure of the telomere length distribution—such as interquartile telomere lengths, median, mean, and the fraction of telomeres of varying length—contains information about the underlying telomere maintenance mechanism of the cell. These observations highlight the potential of DTM to distinguish healthy aging from disease, as well as a tool for clinical investigation.

## 6. Molecular Mechanisms in Aging and Longevity: *Klotho*, ACE, and NF-κΒ

### 6.1. The Klotho Gene

The *Klotho* gene encodes KL, a type-I membrane protein that is primarily expressed in the kidney, brain, and parathyroid gland. *Klotho* functions as an aging-suppressor gene; overexpression leads to extended lifespan, while its deficiency results in accelerated aging phenotypes, thus named accordingly for one of the Greek goddesses of destiny who spun the thread of life [84]. The Klotho protein can be cleaved and released into the circulation, where it acts as a hormone influencing multiple physiological processes. KL modulates the insulin/IGF-1 signaling pathway, which is a key regulator of aging. It inhibits insulin and IGF-1 signaling by decreasing the phosphorylation of insulin receptor substrates, thereby reducing cellular glucose uptake and metabolic activity [85]. This action is thought to mimic the effects of caloric restriction, a well-known intervention that extends lifespan in various organisms.

In addition to its role in regulating glucose metabolism, KL also influences calcium and phosphate homeostasis by acting as a co-receptor for fibroblast growth factor 23 (FGF23). This interaction is crucial for maintaining mineral balance and preventing vascular calcification, a common feature of aging and chronic kidney disease [86]. Recent studies have shown that *Klotho* exhibits protective effects against oxidative stress by upregulating the expression of antioxidant enzymes such as manganese superoxide dismutase (MnSOD) and catalase [87]. This antioxidant function is particularly important in mitigating the damaging effects of reactive oxygen species (ROS), which accumulate with age and contribute to cellular damage. The therapeutic potential of *Klotho* has gained significant attention, with preclinical studies demonstrating that administration of recombinant KL protein can improve cognitive function, reduce vascular calcification, and extend lifespan in animal models [88]. These findings suggest that *Klotho* supplementation or upregulation could serve as a novel intervention for promoting healthy aging and combating age-related diseases.

### 6.2. Angiotensin-Converting Enzyme (ACE) and Aging

The *acn-1* gene is defined as a homolog of nematode ACE, and it has been used to evaluate the relation between the use of *ACE* inhibitors and longevity [89]. The application of the ACE inhibitor, Captopril, reduces the acn-1 activity, leading to life extension, increased stress resistance, and a delay in age-related degenerative changes measured by pharyngeal pumping [90]. Models in mice have also been used to study the effect of ACE inhibitors on lifespan. A study that used Ramipril, the combined application of statin, simvastatin, and ACE inhibitor, resulted in an increased mean lifetime for a group of isocalorically fed mice subjects [91].

Increased resistance to stress has been proven as a factor of life extension in several species, such as humans [92]. As mentioned, ACE is an angiotensin-converting enzyme, and it has been used as a drug target to control hypertension and associated complications. Also, ACE is an amyloid-degrading enzyme, and it has been used to minimize the advanced stage accumulation of beta-amyloid. The different activities of ACE support the notion of aging stages from the transition state to the more advanced pathological state [90]. At this phase of the lifespan, the double ACE functions as an angiotensin-converting enzyme is beneficial to the health status, while the function as an amyloid-degrading enzyme is more important when an abnormal accumulation of beta-amyloid starts [90].

### 6.3. NF-κB and the Inflammatory Response in Aging

The NF-κB (nuclear factor kappa-light-chain-enhancer of activated B cells) pathway is a key regulator of inflammation, immune responses, and cellular senescence, playing a pivotal role in the aging process. Typically inactive in the cytoplasm, NF-κB becomes activated in response to stressors like DNA damage, oxidative stress, and cytokines, translocating to the nucleus to promote the expression of pro-inflammatory genes [93,94]. As cells age, NF-κB activity increases, driving the secretion of the senescence-associated secretory phenotype (SASP)—a collection of pro-inflammatory cytokines, chemokines, and growth factors that contribute to tissue dysfunction and chronic inflammation, or “inflammaging” [40]. This chronic activation leads to age-related diseases like osteoarthritis, cardiovascular disease, and neurodegeneration.

Inhibition of NF-κB has shown promise in reducing cellular senescence and age-related inflammation. For instance, the small molecule SR12343 can block NF-κB activation, extending lifespan and health span in murine models [95]. Natural compounds, such as curcumin, resveratrol, and EGCG, also suppress NF-κB signaling, suggesting potential dietary interventions for aging [96]. NF-κB’s role is evident in specific age-related diseases: It contributes to neuroinflammation in Alzheimer’s disease, vascular inflammation in cardiovascular diseases, and insulin resistance in metabolic disorders [97,98]. Targeting NF-κB offers a promising strategy for reducing inflammaging and delaying age-related pathologies, potentially improving health span and longevity. As research progresses, modulating NF-κB could be a key approach to combating the detrimental effects of aging.

## 7. Other Mechanisms, Pathways, and Factors Underlying Aging

### 7.1. Insulin/IGF-1 Pathway

The insulin/IGF-1 signaling pathway (IIS) regulates aging by controlling processes such as protein synthesis, glucose metabolism, cell proliferation, oxidative stress, and inflammation. The role of IIS in longevity was first discovered in *C. elegans* when the deletion of the *daf-2* gene, which encodes the insulin/IGF-1 receptor, resulted in doubled lifespan. Similar findings in dwarf mouse models suggest a conserved mechanism of action. IGF-1, essential during development, declines with age, but its continued activation contributes to aging by stimulating pathways such as FOXO, NF-κB, and MAPK. IGF-1 also regulates autophagy through Nrf2/Sirt3 signaling. Although reduced IGF-1 signaling can promote longevity, its deficiency leads to sensory hearing loss and increased inflammation, highlighting its complex role in aging. In humans, lower IGF-1 levels with age correlate with worsening hearing loss, suggesting that IGF-1 signaling modulation requires careful consideration in age-related conditions like age-related hearing loss (ARHL) [99,100,101,102].

### 7.2. mTOR Pathway

mTOR kinase is a central regulator of cell growth and metabolism, forming two distinct complexes: mTORC1 and mTORC2. mTORC1 responds to nutrient availability, growth factors, and energy status to promote protein synthesis, lipid biosynthesis, and cell proliferation, while inhibiting autophagy. However, chronic mTORC1 activation has been associated with accelerated aging and age-related pathologies. Inhibition of mTORC1, through caloric restriction or pharmacological agents like rapamycin, has been shown to extend lifespan in various organisms. This extension is attributed to enhanced autophagy, improved protein homeostasis, and reduced cellular senescence. Rapamycin, specifically, has demonstrated efficacy in delaying age-related diseases, including neurodegenerative disorders and cardiovascular diseases, by modulating mTOR activity [103,104,105].

### 7.3. AMPK Pathway

AMPK is a crucial energy sensor activated in response to increased AMP/ATP and ADP/ATP ratios, indicating cellular energy deficits. While AMPK is activated under energy-low conditions, leading to autophagy induction, mTORC1 activity depends on diverse positive signals such as high energy levels, normoxia, amino acids, or growth factors that all result in the inhibition of autophagy [106]. Upon activation, AMPK phosphorylates targets that enhance catabolic pathways, generating ATP, such as glucose uptake and fatty acid oxidation, while inhibiting anabolic processes like lipid and protein synthesis that consume ATP. This dual action restores energy balance and promotes cellular survival under stress conditions. In the context of aging, AMPK activation has been linked to several beneficial effects. It stimulates autophagy by phosphorylating ULK1, initiating the autophagic process essential for removing damaged organelles and proteins, thereby maintaining cellular integrity. Additionally, AMPK enhances mitochondrial biogenesis through the activation of peroxisome proliferator-activated receptor-gamma coactivator 1-alpha (PGC-1α), improving mitochondrial function and reducing reactive oxygen species (ROS) production. These actions collectively contribute to increased health span and delayed onset of age-related diseases [105,107,108,109,110].

### 7.4. Sirtuins

Sirtuins are a family of NAD^+^-dependent deacetylases and ADP-ribosyltransferases involved in regulating metabolic homeostasis, stress responses, and genomic stability. Among the seven mammalian sirtuins (SIRT1–7), SIRT1 and SIRT3 are particularly notable for their roles in aging. SIRT1 deacetylates various substrates, including PGC-1α, FOXO transcription factors, and NF-κB, leading to enhanced mitochondrial function, increased antioxidant defense, and reduced inflammation. SIRT3, localized in mitochondria, deacetylates and activates enzymes involved in oxidative metabolism and antioxidant defense, thereby reducing ROS production and oxidative damage. Activation of sirtuins has been associated with extended lifespan and improved health span in model organisms, suggesting their potential as therapeutic targets for age-related diseases [111,112,113].

### 7.5. Lifestyle Factors

Lifestyle factors significantly influence the activity of these metabolic pathways. Adequate sleep is vital for metabolic health, with sleep deprivation linked to impaired glucose metabolism, increased appetite, and weight gain. Chronic sleep deprivation can disrupt hormonal balance, leading to insulin resistance and increased risk of metabolic syndrome. Moreover, poor sleep quality has been associated with accelerated epigenetic aging, indicating a direct link between sleep and the aging process. Stress management is equally important, as chronic stress activates the hypothalamic–pituitary–adrenal (HPA) axis, resulting in elevated cortisol levels. Prolonged cortisol exposure can impair immune function, increase visceral fat accumulation, and promote a pro-inflammatory state, all of which contribute to accelerated aging. Stress has also been shown to affect the AMPK and mTOR pathways, further linking it to cellular aging processes.

In summary, the interplay between key metabolic pathways and lifestyle factors plays a pivotal role in regulating aging and longevity. Understanding these relationships offers valuable insights into potential interventions aimed at promoting health span and delaying the onset of age-associated diseases [114,115,116,117,118].

### 7.6. Gut Microbiome

The gut microbiome, comprising trillions of microorganisms residing in the gastrointestinal tract, plays a pivotal role in human health. Emerging evidence indicates that age-associated alterations in gut microbiota composition significantly influence systemic inflammation and the aging process. Understanding these changes and exploring targeted interventions may offer promising avenues for promoting healthy aging [119].

*Age-Associated Changes in Gut Microbiota Composition:* Throughout the human lifespan, the gut microbiota undergoes dynamic shifts. In infancy, the gut is predominantly colonized by the *Bacillota*/*Bacteroidota* phyla, which decline after weaning. As individuals age, there is a notable decrease in microbial diversity, with reductions in beneficial taxa such as *Bifidobacterium* and increases in potentially pathogenic bacteria. Studies have reported that older adults exhibit distinct gut microbiota profiles compared to younger individuals, characterized by a decrease in *Firmicutes* and an increase in *Bacteroidetes*. These compositional changes are associated with various health outcomes, including increased frailty and inflammation [26,120].

*Impact of the Microbiome on Systemic Inflammation:* The gut microbiome significantly influences systemic inflammation, a key factor in the aging process. Age-related dysbiosis can compromise the integrity of the intestinal barrier, leading to increased permeability and translocation of microbial products such as lipopolysaccharides into the bloodstream. This translocation triggers chronic low-grade inflammation, commonly referred to as “inflammaging,” which is implicated in the pathogenesis of various age-related diseases, including cardiovascular disease and neurodegeneration [18,19].

*Potential Interventions Targeting the Microbiome to Promote Healthy Aging:* Modulating the gut microbiome presents a promising strategy for promoting healthy aging. Potential interventions include the following: Dietary Modifications: Incorporating prebiotic and probiotic-rich foods can enhance beneficial microbial populations. Diets high in fiber support the growth of short-chain fatty acid-producing bacteria, which have anti-inflammatory properties. Conversely, reducing the intake of processed foods and saturated fats may prevent dysbiosis. Probiotic Supplementation: Administering specific probiotic strains has been shown to restore microbial balance, improve gut barrier function, and reduce systemic inflammation. Clinical studies have demonstrated that probiotics can aid in managing conditions like irritable bowel syndrome and may have potential benefits in reducing inflammation associated with aging [119,121].

*Fecal Microbiota Transplantation (FMT):* FMT involves the transfer of fecal material from a healthy donor to the gastrointestinal tract of a recipient. Emerging research suggests that FMT can rejuvenate the gut microbiome, enhance immune function, and reduce inflammation in older adults. However, further studies are needed to establish its efficacy and safety in the context of aging [122].

*Lifestyle Interventions:* Regular physical activity and stress management have been associated with favorable shifts in gut microbiota composition. Exercise can increase microbial diversity, while stress reduction techniques may mitigate stress-induced dysbiosis, thereby supporting overall health during aging [123].

In summary, age-related changes in gut microbiota composition contribute to systemic inflammation and influence the aging process. Targeted interventions aimed at modulating the microbiome hold potential for promoting healthy aging and mitigating age-associated diseases. Ongoing research is essential to elucidate the mechanisms underlying these interactions and to develop effective microbiome-based therapies.

## 8. Epigenetic Alterations

Aging is a complex biological process characterized by progressive functional decline at cellular and organismal levels. Emerging evidence highlights the pivotal role of epigenetic alterations, including DNA methylation, histone modifications, stem cell exhaustion, proteostasis imbalance, mitochondrial dysfunction, and disrupted intercellular communication, in driving aging and age-related diseases.

Aging is accompanied by global hypomethylation and regional hypermethylation, contributing to genomic instability and altered gene expression. Epigenetic clocks, particularly the Horvath clock, estimate biological age using DNA methylation patterns, correlating with age-related diseases and mortality [124]. Horvarth clock is an algorithm to extract DNA methylation sites and predict biological age. This epigenetic clock is based on 353 DNA CpG methylation sites that together form an aging clock in terms of chromatin states and tissue variance and can accurately determine the biological age in different cells and tissues [125]. Thus, biomarkers of aging based on DNA methylation data enable accurate age estimates for any tissue across the entire life course [124]. Age-related DNA methylation changes, including hypermethylation at specific CpG islands and hypomethylation in repetitive sequences, influence gene silencing and genomic instability [126,127]. These changes are not random but are linked to biological aging mechanisms and potential longevity pathways. However, most age-related differentially methylated regions (a-DMRs) appear neutral concerning biological age markers [128].

Apart from DNA methylation patterns, covalent modifications of its closely bound partners, histones, are instrumental in gene expression with age. Thus, histone modifications, such as acetylation and methylation, significantly influence chromatin structure and gene expression during aging. Notably, aging is associated with increased H3K9ac and decreased H3K27me3, contributing to cellular senescence and genomic instability [129]. In addition, comparison of H3K4me3 and H3K27me3 levels in proliferating and senescent human fibroblasts revealed large-scale chromatin modifications during replicative senescence. Thus, in senescent cells, H3K4me3 and H3K27me3 are high and frequently colocalize in areas extended to hundreds of kilobases called “mesas”, in contrast to “canyons” (domains up to 10 Mb) that mostly form between lamin B1 (LMNB1)-associated domains; they are enriched in genes and enhancers, and the levels of H3K27me3 are low. H3K4me3 and H3K27me3 mesas colocalize in domains that associate with LMNB1 and overlap DNA hypomethylation, while canyons are enriched in gene bodies and enhancers. During senescence, loss of H3K27me3 has been associated with the upregulation of senescent transcriptional programs chromatin reorganization and lamin B1 downregulation is central to global and local chromatin changes that impact gene expression, aging, and cancer [130]. Histone acetylation patterns influence DNA damage repair and age-related gene expression, with histone deacetylases linked to longevity and genomic stability [131,132]. Redistribution of histone modifications over time affects gene regulation and inflammatory responses [133]. Interventions such as caloric restriction have been shown to slow histone modification changes, suggesting lifestyle factors can modulate epigenetic aging [134].

Aging impairs stem cell function due to cumulative genetic and epigenetic alterations, reducing regenerative capacity and increasing disease susceptibility. Epigenetic reprogramming influences “epithelial memory,” enhancing cellular transformation during inflammation and affecting cancer progression [135]. Additionally, wound memory progenitors exhibit primed chromatin states, promoting efficient wound repair but increasing cancer risk [136]. Single-cell epigenomic analyses reveal age-related changes in stem cell populations, providing insights into tissue-specific aging dynamics.

## 9. Proteostasis and Mitochondrial Dysfunction

Loss of proteostasis, characterized by protein misfolding and aggregation, is a hallmark of aging and neurodegenerative diseases. Small-molecule metabolites influence proteostasis and link cellular metabolic states with aging mechanisms [9]. Proteostatic imbalance leads to proteinaceous aggregates, contributing to cellular dysfunction in neurodegenerative disorders such as Alzheimer’s disease [137].

Mitochondrial dysfunction is a critical driver of aging, characterized by mtDNA mutations, oxidative stress, and impaired energy metabolism. In aging, increased expression of *NOX4* encoding for NADPH oxidase 4 induces mitochondrial dysfunction and a pro-inflammatory phenotype, advancing atherosclerosis [12]. Targeting mitochondrial function presents a promising strategy to alleviate age-related diseases and promote healthy aging [11].

Aging disrupts intercellular communication pathways, contributing to chronic inflammation and tissue dysfunction. Changes in cellular signaling, including chemokines, cytokines, and endocrine factors, affect tissue homeostasis and age-related disease progression [138]. Immortalized cell models provide insights into fundamental communication patterns, advancing the understanding of aging-associated cellular interactions [139].

## 10. Current and Emerging Anti-Aging Strategies

Advancements in biomedical research have led to the development of various interventions aimed at mitigating the effects of aging and extending healthy lifespan. This section provides an overview of current and emerging anti-aging strategies, including senolytics, telomerase activation, gene therapy, personalized medicine, and pharmacological agents.

As mentioned above, cellular senescence is characterized by irreversible cell cycle arrest and contributes to aging and age-related diseases through the accumulation of senescent cells and their pro-inflammatory secretions. Senolytics are a class of drugs designed to selectively eliminate these senescent cells, thereby alleviating their detrimental effects. Preclinical studies have demonstrated that senolytics can improve tissue function and extend lifespan in animal models. For instance, the combination of dasatinib and quercetin has been shown to reduce senescent cell burden and ameliorate age-related pathologies. Ongoing clinical trials are evaluating the safety and efficacy of senolytic therapies in humans [140,141,142].

Strategies to activate telomerase include small-molecule activators and gene therapy approaches. Resveratrol, a natural polyphenol, has been identified as a telomerase activator with potential anti-aging effects. Gene therapy delivering the *TERT* gene has shown promise in extending lifespan in mouse models. However, concerns regarding the potential for increased cancer risk due to uncontrolled cell proliferation necessitate cautious advancement in this area [143,144].

Gene therapy offers the potential to directly modify genetic factors associated with aging. Approaches include the delivery of genes that promote cellular rejuvenation or the suppression of genes that contribute to aging processes. For example, upregulating the expression of the *Klotho* gene has been associated with extended lifespan in animal studies. Gene therapies targeting aging-related genes are explored, though they are still in experimental stages and require extensive validation [145].

Personalized medicine tailors healthcare interventions based on individual genetic, environmental, and lifestyle factors. In the context of aging, this approach involves assessing the biomarkers of aging and implementing customized strategies to mitigate age-related decline. Advancements in genomics, proteomics, and metabolomics facilitate the identification of individual aging profiles, enabling interventions such as personalized nutrition plans, targeted pharmacotherapy, and lifestyle modifications to promote healthy aging [146,147,148,149].

Several pharmacological agents are under investigation for their potential anti-aging effects. Rapamycin is an mTOR inhibitor that has been shown to extend lifespan in various species by modulating pathways involved in cell growth and metabolism [147]. Metformin, a widely used antidiabetic drug, has demonstrated potential in extending lifespan and delaying age-related diseases in preclinical studies. NAD^+^ precursors are compounds, such as nicotinamide riboside, that aim to boost cellular levels of NAD^+^, a coenzyme involved in metabolic processes, which decline with age. While these agents show promise, rigorous clinical trials are necessary to establish their efficacy and safety in humans. In summary, a multifaceted array of interventions targeting the fundamental mechanisms of aging is under active investigation. Continued research is essential to translate these strategies into safe and effective therapies for promoting healthy aging in humans.

## 11. Emerging and Underexplored Signaling Pathways in Aging: Insights from Systems Biology and Omics

Recent systems biology and multi-omics approaches have expanded our understanding of aging beyond classical pathways, revealing novel regulatory networks with potential roles in cellular and organismal senescence. High-throughput transcriptomics, proteomics, metabolomics, and single-cell sequencing have identified several emerging players that modulate age-associated processes.

Furthermore, recent data suggest roles for RNA-binding proteins and RNA methylation regulators, such as m6A writers and erasers (e.g., METTL3 m6A-methyltransferase and FTO demethylase), in controlling the stability and translation of transcripts involved in inflammation, metabolism, and senescence [124]. This epitranscriptomic regulation may serve as a fine-tuning mechanism in the adaptive stress response during aging.

Finally, non-coding RNAs, including microRNAs and long non-coding RNAs (lncRNAs), are increasingly recognized as master regulators of aging phenotypes, with roles in chromatin remodeling, SASP modulation, and telomere maintenance [13,40]. These insights offer new potential targets for aging-related therapeutic interventions and highlight the value of integrative approaches in unraveling the complexity of the aging process.

## 12. Conclusions

The study of aging is increasingly revealing the extensive interconnectedness of molecular pathways, many of which were previously considered in isolation. In particular, the dynamic interplay between AMPK, mTOR, and sirtuin signaling emerges as a central regulatory hub that integrates nutrient sensing, mitochondrial function, and cellular stress responses. These pathways collectively influence cellular fate decisions, such as autophagy, senescence, and metabolic reprogramming—thereby impacting tissue homeostasis and organismal aging.

Recent studies have also illuminated the cross-talk between metabolic regulators and immune aging, or immunosenescence, highlighting a feedback loop in which age-related metabolic shifts exacerbate chronic inflammation, while inflammatory mediators in turn modulate energy sensing and mitochondrial activity. For instance, AMPK activation has been shown to modulate the SASP profile, and mTOR inhibition can blunt inflammatory signaling, suggesting potential synergies in targeting these pathways therapeutically.

Moreover, the epitranscriptomic landscape, especially RNA modifications such as N6-methyladenosine (m6A), adds an additional layer of regulation by fine-tuning RNA stability, splicing, and translation in response to cellular stress and inflammation. These modifications have recently been implicated in the regulation of both senescence and immune function, pointing to novel mechanisms of age-related transcriptomic remodeling.

Finally, telomere dynamics and key regulatory genes, such as *Klotho*, ACE, and NF-κB, have provided invaluable insights into the molecular mechanisms underlying aging and longevity. Telomere shortening, inflammation, oxidative stress, and dysregulated autophagy contribute to the aging process and the development of age-related diseases. Understanding these pathways offers promising avenues for developing interventions that target the fundamental aspects of aging. Future research should focus on elucidating the intricate networks governing cellular senescence, exploring the potential of telomerase activators, *Klotho* enhancers, and NF-κB inhibitors, and translating these findings into clinical therapies. By addressing the root causes of aging, it may be possible to extend not only lifespan but also health span, ultimately improving the quality of life for the aging population.

## Figures and Tables

**Table 2 genes-16-00796-t002:** Sources of genomic instability.

DNA damage
2. Defective DNA repair
3. Replication stress
4. DNA replication errors
5. Chromosomal abnormalities
6. Chromosomal rearrangements
7. Genome rearrangement events
8. Telomere attrition
9. Oncogene activation
10. Epigenetic alterations
11. Genotoxic agents
12. Mitotic errors

## Data Availability

No new data were created or analyzed in this study.

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
