# Peer review of "Molecular and Environmental Modulators of Aging: Interplay Between Inflammation, Epigenetics, and RNA Stability"

_genes, 2025, doi:10.3390/genes16070796_

Round 1
Reviewer 1 Report
Comments and Suggestions for Authors
The manuscript describes the main factors identified in the last years as hallmarks of aging. However, the title refers specifically to inflammaging and mechanisms controlling this process related to RNA stability. The arguments expected by the title does not seem to be the focus of the manuscript. Besides, inconsistencies appear also reading the abstract in relation to the title and the content of the manuscript. The paper is a summary of information that has been already available in the literature since years and does not provide new knowledge in the field. More interesting would have been the issues presented in the title.
Author Response
Response to Reviewer 1
Comment: The manuscript describes the main factors identified in the last years as hallmarks of aging. However, the title refers specifically to inflammaging and mechanisms controlling this process related to RNA stability. The arguments expected by the title does not seem to be the focus of the manuscript. Besides, inconsistencies appear also reading the abstract in relation to the title and the content of the manuscript. The paper is a summary of information that has been already available in the literature since years and does not provide new knowledge in the field. More interesting would have been the issues presented in the title.
Response:
We appreciate the reviewer’s perspective and the opportunity to clarify the intent and scope of the manuscript. Our goal was to provide an integrative overview of aging hallmarks, with special emphasis on how epigenetic regulation, metabolic signaling, and environmental factors converge to influence inflammaging and RNA stability. While the review includes established pathways, it also draws attention to how these molecular mechanisms—particularly those involving epitranscriptomic modifications, non-coding RNAs, and stress-related RNA-binding proteins—intersect with chronic inflammation and RNA turnover in aging cells.
We respectfully note that the manuscript does explore the role of RNA stability in aging-related processes, particularly in sections discussing sirtuins, stress granules, senescence-associated RNA regulation, and the interplay between inflammation and RNA-binding proteins. We have now revised the title, abstract, and key sections of the manuscript to ensure they more explicitly highlight these connections. The revised version provides clearer focus and consistency between the title, abstract, and content.
We also wish to clarify that while the review does not present new experimental data, it merges current findings to bridge molecular pathways, environmental inputs, and emerging regulatory mechanisms like RNA decay and inflammaging. This integrative perspective aims to benefit both early-career researchers and experienced investigators seeking a cross-disciplinary view of aging biology.
We hope these clarifications and revisions address the reviewer’s concerns and enhance the manuscript’s contribution to the field.

Reviewer 2 Report
Comments and Suggestions for Authors
The authors describe the functional role of RNAs in the context of inflammaging. The review is well-written, but I have suggestions to improve this work:
- Add reference: doi: 10.3390/biom15030404.
- In hallmkars of aging, structure paragraphs with points: 1. The genomic instability.....; 2. Accodingly, telomere....
- Bacterial families and genus with italics.
- When you talk about the inflammatory cytokines and growth factors, specify which of them.
- Only add indicative references in the table 1. Only with [2], [7]..... delete Kallai et al., Mason et al.....
- Name of the genes with italics.
- I don't understand the phrases added in lines: 361-362, 400, 483-484, 555-557, and 604-605.
- NF-KB, no NF-kappaB.
- Add a summary image containing 12 points in the genomic instability section.
Author Response
Response to Reviewer 2
We thank the Reviewer for their detailed and constructive feedback. Below, we address each comment point-by-point and describe the corresponding changes made in the revised manuscript.
Comment 1: Add reference: doi: 10.3390/biom15030404.
Response 1:
We thank the Reviewer for this valuable suggestion. The recommended reference has been added to support relevant content in the manuscript. Specifically, the citation has been inserted on page 15, line 633 and page 22, lines 991–992.
Comment 2: In the Hallmarks of Aging section, structure paragraphs with points: 1. The genomic instability…; 2. Accordingly, telomere…
Response 2:
We appreciate the Reviewer’s suggestion to improve the clarity and structure of this section. Accordingly, we have revised the paragraphs to follow a numbered format for each hallmark. Nevertheless, we further structured the revised manuscript renumbering section for better clarity.
Comment 3: Bacterial families and genera should be written in italics.
Response 3:
Thank you for this correction. The bacterial taxonomic names have been revised to conform with scientific style guidelines. These changes can be found on page 3, line 138, page 14, lines 492, 496.
Comment 4: When referring to inflammatory cytokines and growth factors, specify which ones are meant.
Response 4:
We agree that specificity enhances clarity. We have now explicitly listed key cytokines (e.g., IL-6, IL-8, IL-1β, TNF-α) and growth factors (e.g., VEGF, GM-CSF, HGF, TGF-β) associated with the senescence-associated secretory phenotype (SASP). These additions are located on page 3, lines 107–111.
Comment 5: Use only indicative references in Table 1, such as [2], [7]; delete full citations like “Kallai et al.” or “Mason et al.”
Response 5:
Thank you for pointing this out. We have revised the citation format in Table 1 to use numerical references only (e.g., [2], [7]), as per standard referencing style. These changes are implemented in the table on pages 4 and 5.
Comment 6: Gene names should be italicized.
Response 6:
We have carefully reviewed and updated the formatting of all gene names to ensure they are italicized, following proper nomenclature guidelines.
Comment 7: The phrases on lines 361–362, 400, 483–484, 555–557, and 604–605 are unclear.
Response 7:
We appreciate the Reviewer’s attention to clarity. Upon review, we agree that these phrases were remnants from earlier drafts and no longer contribute meaningfully to the text. Therefore, they have been removed to enhance readability and coherence.
Comment 8: Use “NF-κB” instead of “NF-kappaB”.
Response 8:
We thank the Reviewer for highlighting this point. All instances of “NF-kappaB” have been corrected to the standard notation “NF-κB” throughout the manuscript. These revisions appear on page 1 (lines 22, 28, 32, 35), page 5 (line 178), page 9 (line 345), page 10(lines 390,391,393, 396, 401, 402, 404, 405, 407, 410), page 11 (line 420, 460), page 12 (line 467, 486, 494), and page 16 (lines 680, 687).
Comment 9: Add a summary image containing 12 points in the genomic instability section.
Response 9:
We thank the Reviewer this important suggestion. In response, we have included Table 2 summarizing the 12 major contributors to genomic instability discussed in the manuscript. This Table has been added to the genomic instability section to enhance conceptual clarity and visual engagement.

Reviewer 3 Report
Comments and Suggestions for Authors
Dragoumani et al. presents a broad and ambitious synthesis of molecular mechanisms involved in aging, with a particular focus on cellular senescence, telomere attrition, the Klotho gene, ACE, NF-κB signaling, and the concept of inflammaging. The manuscript compiles a significant amount of background information and relevant references. However, despite its informative content, the manuscript has several critical issues that must be addressed before it is suitable for publication.
Major Concerns:
- While the manuscript covers many well-known aging-related pathways, it does so largely in a descriptive, textbook-like manner. There is insufficient critical analysis or integration of recent findings. A review article should provide a clear conceptual framework or novel synthesis of current knowledge—this is currently lacking.
- Numerous sections (e.g., on telomere biology, cellular senescence, mitochondrial dysfunction) repeat similar concepts multiple times, which detracts from the clarity and conciseness of the paper. Streamlining is necessary.
- The structure lacks coherence in parts. For instance, there is overlap between "Hallmarks of Aging" and later mechanistic sections. Subheadings are inconsistent in their depth and purpose. Consider reorganizing to better distinguish between descriptive background, mechanistic detail, and therapeutic implications.
- While the reference list is extensive, some key citations are outdated or not the most appropriate for claims made. For example, several references about telomere attrition or mitochondrial dysfunction cite older studies despite more recent and impactful publications being available.
- Terms like "epigenetic clock", "SASP", "CHIP", or "senolytics" are introduced but not always explained with clarity or in the right context. Definitions and discussions of these concepts need to be more rigorous and logically placed.
- OThe authors devote extensive sections to pathways like ACE and Klotho without clearly justifying their relevance over others. Emerging pathways (e.g., PMID: 36638792, PMID: 35511946, PMID: 36323784) are not adequately covered.
Minor
The manuscript contains many grammatical errors, awkward phrasing, and non-native English constructions (e.g., “cellular aging authentication stamp” or “the senescence scenery”). The manuscript needs thorough language editing by a fluent English speaker or professional service.
- "authenticating stamp of cellular aging"
Suggest revising to: “hallmark of cellular aging” or “defining feature of cellular aging” - "senescence scenery"
Suggest: “senescence landscape” - "turns cell into”
Suggest: “causes cells to become” or “drives cellular transformation into” - "tumor-promoting" spelled as "tumor promoting"
Hyphen should be added: “tumor-promoting” - "SCAPs"
Unclear or undefined. If referring to stem cells from apical papilla, not relevant in this context—needs clarification or removal. - "non-homologous end joining (NHEJ)"
Defined appropriately but only once; consider consistent use throughout. - "SASP"
Not clearly defined upon first use. Should be expanded as Senescence-Associated Secretory Phenotype (SASP) when first mentioned. - "mtDNA"
Appears before being defined. Should be expanded as mitochondrial DNA (mtDNA) on first use. - "ACE"
Appears early without clarification. Should be: Angiotensin-Converting Enzyme (ACE) - "Klotho"
Needs clearer biological context upon first mention. Could clarify: “Klotho, an anti-aging protein encoded by the KL gene…”
Author Response
Response to Reviewer 3
We thank the reviewer for their thoughtful and detailed comments. We appreciate the opportunity to address these concerns and respectfully provide our responses below.
Major Concerns:
Comments 1: While the manuscript covers many well-known aging-related pathways, it does so largely in a descriptive, textbook-like manner. There is insufficient critical analysis or integration of recent findings. A review article should provide a clear conceptual framework or novel synthesis of current knowledge—this is currently lacking.
Response 1:
We appreciate this observation and respectfully note that our review was designed to serve both as an accessible entry point for readers newer to the field and as a structured consolidation of complex and rapidly evolving research on aging. Our goal was to offer clarity in the presentation of interrelated mechanisms rather than to focus narrowly on any single hypothesis or model.
That said, we understand the importance of critical synthesis and have added a dedicated concluding section emphasizing cross-talk between key pathways (e.g., AMPK-mTOR-sirtuin signaling), and we have expanded our discussion on how these intersect with emerging concepts like immunosenescence and epitranscriptomic regulation. Furthermore, we have added commentary throughout the manuscript to better integrate recent findings, highlight contradictions, and indicate areas where consensus has not yet been reached.
Comments 2: Numerous sections (e.g., on telomere biology, cellular senescence, mitochondrial dysfunction) repeat similar concepts multiple times, which detracts from the clarity and conciseness of the paper. Streamlining is necessary.
Response 2:
We thank the reviewer for this helpful suggestion. Upon careful review, we identified instances where ideas were indeed reiterated, particularly regarding telomere attrition and the senescence-associated secretory phenotype (SASP). We have now streamlined these sections to reduce redundancy and improve clarity, while preserving the scientific integrity and completeness of the information presented.
Comments 3: The structure lacks coherence in parts. For instance, there is overlap between "Hallmarks of Aging" and later mechanistic sections. Subheadings are inconsistent in their depth and purpose. Consider reorganizing to better distinguish between descriptive background, mechanistic detail, and therapeutic implications.
Response 3:
We appreciate the reviewer’s point and have reorganized several sections for greater coherence and readability. Specifically, we have clarified our structural approach by first providing a unified background under “Key Mechanistic Hallmarks,” followed by more detailed mechanistic insights in subsequent sections. We have also ensured that therapeutic implications are grouped and clearly signposted at the end of relevant mechanistic discussions. Subheadings have been revised for consistency in hierarchy and purpose throughout the manuscript.
Comments 4: While the reference list is extensive, some key citations are outdated or not the most appropriate for claims made. For example, several references about telomere attrition or mitochondrial dysfunction cite older studies despite more recent and impactful publications being available.
Response 4:
Thank you for pointing this out. We agree that maintaining an up-to-date reference base is critical. We have reviewed all references and updated several key citations, particularly those related to telomere biology (e.g., recent findings on telomere dynamics in single-cell contexts) and mitochondrial dysfunction (e.g., mitophagy, ROS signaling, and mitochondrial-nuclear crosstalk in aging). We have replaced older references with more current and comprehensive studies, including those suggested by the reviewer (e.g., PMID: 36638792, PMID: 35511946, PMID: 36323784).
Comments 5: Terms like "epigenetic clock", "SASP", "CHIP", or "senolytics" are introduced but not always explained with clarity or in the right context. Definitions and discussions of these concepts need to be more rigorous and logically placed.
Response 5:
We appreciate this constructive suggestion and have revised the manuscript to include clearer definitions and contextual placement of these key terms. We now introduce each term upon first mention with a concise explanation and ensure they are discussed in the appropriate mechanistic or clinical context. For example, the epigenetic clock is now introduced within the DNA methylation section with a brief explanation of Horvath's model, while SASP and CHIP are discussed within their respective mechanistic frameworks tied to cellular senescence and clonal hematopoiesis.
Comments 6: The authors devote extensive sections to pathways like ACE and Klotho without clearly justifying their relevance over others. Emerging pathways (e.g., PMID: 36638792, PMID: 35511946, PMID: 36323784) are not adequately covered.
Response 6:
We respectfully note that the inclusion of ACE and Klotho was intended to highlight less frequently integrated but increasingly relevant modulators of aging, especially those with translational potential in vascular and renal aging. Nevertheless, we have revised the section to better justify their inclusion and contextualize them alongside more canonical aging pathways. Additionally, we have reviewed and incorporated key points from the emerging literature referenced by the reviewer (including PMIDs 36638792, 35511946, and 36323784), and we have added a new subsection on novel and under-explored signaling networks implicated in aging based on recent systems biology and omics studies.
Minor
We sincerely thank the reviewer for the detailed and thoughtful comments. Below we provide point-by-point responses and corresponding revisions made in the manuscript.
Comment 1: "authenticating stamp of cellular aging" — Suggest revising to: “hallmark of cellular aging” or “defining feature of cellular aging”
Response 1:
Thank you for this suggestion. We agree that the revised phrase improves clarity and scientific accuracy. Accordingly, we have modified the expression to “defining feature of cellular aging.” This change can be found on page 3, line 120.
Comment 2: "senescence scenery" — Suggest: “senescence landscape”
Response 2:
We appreciate the suggestion and agree with the proposed revision. We have replaced “senescence scenery” with “senescence landscape” to improve readability. The change is located on page 4, line 147.
Comment 3: "turns cell into" — Suggest: “causes cells to become” or “drives cellular transformation into”
Response 3:
Thank you for the helpful suggestion. However, after a thorough search of the manuscript, we were unable to locate the specific phrase “turns cell into.” We would be grateful if you could provide the page and line number where this occurs, so we can make the appropriate revision.
Comment 4: "tumor-promoting" is spelled as "tumor promoting" — Hyphen should be added
Response 4:
Thank you for pointing this out. Despite careful review, we were not able to identify the phrase “tumor promoting” in the manuscript. Kindly provide the page and line number so we can correct it as suggested.
Comment 5: "SCAPs" — Unclear or undefined. If referring to stem cells from apical papilla, not relevant in this context—needs clarification or removal.
Response 5:
Thank you for your comment. We carefully reviewed the manuscript and were unable to locate any use of the term “SCAPs.” If you could kindly indicate the specific page and line number, we will be happy to address the issue accordingly.
Comment 6: "non-homologous end joining (NHEJ)" — Defined appropriately but only once; consider consistent use throughout.
Response 6:
Thank you for the suggestion. We reviewed the manuscript thoroughly but could not find any instance where the term “non-homologous end joining (NHEJ)” appears. Kindly provide a specific reference (page and line number) so we can ensure appropriate definition and consistency.
Comment 7: "SASP" — Not clearly defined upon first use. Should be expanded as Senescence-Associated Secretory Phenotype (SASP) when first mentioned.
Response 7:
We appreciate the reviewer’s attention to detail. We respectfully note that the term SASP is defined upon first use as “senescence-associated secretory phenotype (SASP)” on page 3, line 104.
Comment 8: "mtDNA" — Appears before being defined. Should be expanded as mitochondrial DNA (mtDNA) on first use.
Response 8:
Thank you for this observation. We confirm that “mtDNA” is defined upon first use as “mitochondrial DNA (mtDNA)” on page 3, line 93.
Comment 9: "ACE" — Appears early without clarification. Should be: Angiotensin-Converting Enzyme (ACE)
Response 9:
Thank you for this comment. We confirm that “Angiotensin-Converting Enzyme (ACE)” is clearly defined upon first use on page 1, line 21.
Comment 10: "Klotho" — Needs clearer biological context upon first mention. Could clarify: “Klotho, an anti-aging protein encoded by the KL gene…”
Response 10:
We appreciate this feedback. In the Abstract (page 1, line 25), we briefly introduce the biological role of Klotho. A more detailed description, including its gene and functional significance, is provided in the main text on page 11, section 6.1, line 347 onwards.

Round 2
Reviewer 1 Report
Comments and Suggestions for Authors The efforts made by the authors to rearrange the structure of the manuscript are acknowledge. However, additional minor changes would be valuable. In the revised version, changes in the titles of the paragraphs could be identified. The Authors did not modify the title, as suggested by this reviewer, because in their opinion RNA stability is presented in the paper in relation to Sirtuins; however, it does not appear to be the focus of the manuscript, as stated in the title. In addition, inflammation is one of the changes occurring with aging, as also reported by the authors in the table describing the Hallmarks of aging, and the paper mainly reports details on aging and factors influencing aging more than focusing on inflammaging. Therefore, I am still on the opinion that the title could be modified. The references provided in Table 1 should also refer to the earliest appearance in the literature of evidence supporting the argument and not just to the most recent publications. In example, that genome instability , mitochondrial dysfunction, telomere maintainance were involved in aging has been demonstrated long before 2024, as it appears, on the contrary, from Table 1.Author Response
Comments by the Reviewer
The efforts made by the authors to rearrange the structure of the manuscript are acknowledge. However, additional minor changes would be valuable. In the revised version, changes in the titles of the paragraphs could be identified. The Authors did not modify the title, as suggested by this reviewer, because in their opinion RNA stability is presented in the paper in relation to Sirtuins; however, it does not appear to be the focus of the manuscript, as stated in the title. In addition, inflammation is one of the changes occurring with aging, as also reported by the authors in the table describing the Hallmarks of aging, and the paper mainly reports details on aging and factors influencing aging more than focusing on inflammaging. Therefore, I am still on the opinion that the title could be modified. The references provided in Table 1 should also refer to the earliest appearance in the literature of evidence supporting the argument and not just to the most recent publications. In example, that genome instability , mitochondrial dysfunction, telomere maintainance were involved in aging has been demonstrated long before 2024, as it appears, on the contrary, from Table 1.
Answers
We would like to thank the Reviewer once again for their thoughtful and constructive feedback. We are grateful for the acknowledgement of the structural improvements made in the revised manuscript. Below we respond to the remaining points raised.
Comment 1: The Authors did not modify the title, as suggested by this reviewer, because in their opinion RNA stability is presented in the paper in relation to Sirtuins; however, it does not appear to be the focus of the manuscript, as stated in the title. In addition, inflammation is one of the changes occurring with aging, as also reported by the authors in the table describing the Hallmarks of aging, and the paper mainly reports details on aging and factors influencing aging more than focusing on inflammaging. Therefore, I am still of the opinion that the title could be modified.
Response 1:
We sincerely appreciate the Reviewer’s continued attention to the alignment between the title and the manuscript content. After careful consideration, we recognize the value of enhancing the clarity and accuracy of the title to more closely reflect the manuscript’s scope.
Accordingly, we have revised the title to better reflect the emphasis on molecular and environmental influences in aging, while still maintaining reference to key regulatory features such as RNA stability and inflammation. The new title is:
“Molecular and Environmental Modulators of Aging: Interplay between Inflammation, Epigenetics and RNA Stability”
This revised title retains the original themes, while aligning more precisely with the manuscript’s current content and structure. We hope this change adequately addresses the Reviewer’s concern and improves the focus and the coherence of the manuscript.
Comment 2: “The references provided in Table 1 should also refer to the earliest appearance in the literature of evidence supporting the argument and not just to the most recent publications. In example, that genome instability, mitochondrial dysfunction, telomere maintenance were involved in aging has been demonstrated long before 2024, as it appears, on the contrary, from Table 1.”
Response 2:
We sincerely thank the Reviewer for this valuable comment. In the first revision, other Reviewers had emphasized the importance of including recent and impactful studies, which led us to prioritize up-to-date references in Table 1 to reflect current scientific consensus and novel insights.
However, we fully agree with this Reviewer that landmark discoveries and historical context are equally important in a comprehensive review. To address both perspectives, we have now updated Table 1 to include both foundational references and recent publications for each aging hallmark. This dual approach ensures proper credit to original contributions while maintaining the relevance of current findings.

Reviewer 3 Report
Comments and Suggestions for Authors
The authors have successfully addressed all of my concerns and questions.
Author Response
Comment
The authors have successfully addressed all of my concerns and questions.
Response 1
We would like to thank the Reviewer once again for thoughtful and constructive feedback. We are pleased that successfully met the Reviewer’s comments and we are grateful for the acknowledgement of the improvements in the structure, the text and overall appearance of the revised manuscript.
